# Transformative learning of medical trainees during the COVID-19 pandemic: A mixed methods study

Benjamin Vipler [1,2]*, Bethany Snyder[3], Jennifer McCall-Hosenfeld[2,4], Paul Haidet[2,4,5], Mark Peyrot[6], Heather Stuckey[2,3]

1 Division of Hospital Medicine, University of Colorado Hospital, Aurora, CO, United States of America, 2 Division of General Internal Medicine, Penn State Health Milton S Hershey Medical Center, Hershey, PA, United States of America, 3 Qualitative and Mixed Methods Core, Penn State College of Medicine, Hershey, PA, United States of America, 4 Department of Public Health Sciences, Penn State College of Medicine, Hershey, PA, United States of America, 5 Department of Humanities, Penn State College of Medicine, Hershey, PA, United States of America, 6 Department of Sociology, Loyola University Maryland, Baltimore, MD, United States of America

* benjamin.vipler@cuanschutz.edu

**Data Availability Statement:** All relevant data are within the manuscript and its Supporting Information files.

## Abstract

### Background

The coronavirus disease 2019 (COVID-19) pandemic has had a transformative effect on individuals across the world, including those in healthcare. Transformative learning is an educational theory in which an individual's worldview is fundamentally altered through conscious reflection (Cognitive Rational), insights (Extrarational), or social reform (Social Critique). We utilized transformative learning theory to characterize the experiences of medical trainees during the pandemic.

### Methods

We used the Transformative Learning Survey in September and October 2020 to evaluate the processes and outcomes of transformative learning in health professions students and housestaff at an academic medical center during the pandemic. We analyzed survey scores for three process domains and four outcome subdomains. We inductively coded the survey's two open-ended questions and performed qualitative and mixed-methods analyses.

### Results

The most prominent TL outcome was Self-Awareness, Acting Differently was intermediate, and Openness and Worldview Shifts were lowest. Cognitive Rational and Social Critique processes were more prominent than Extrarational processes. Students were more likely than housestaff to undergo transformative learning through the Social Critique process (p = 0.025), in particular the sub-processes of Social Action (p = 0.023) and Ideology Critique (p = 0.010). Qualitative analysis via the aggregation of codes identified four responses to the pandemic: negative change, positive change, existential change, or no change. Negative changes (67.7%) were most common, with students reporting more of these changes than housestaff (74.8% vs 53.6%; p < 0.01). Only 8.4% of reported changes could be defined as transformative

**Funding:** Funding for this study was awarded (BV) through a Penn State Department of Medicine (https://med.psu.edu/medicine) Inspiration Pilot Grant for COVID-19 Related Research (INSPIREVIP2020). The funder provided support in the form of salaries for authors (BV, JMH, PH, HS), but did not have any additional role in the study design, data collection and analysis, decision to publish, or preparation of the manuscript. The specific roles of these authors are articulated in the 'author contributions' section.

**Competing interests:** The authors have declared that no competing interests exist.

## Conclusions

Through the theoretical lens of transformative learning, our study provides insight into the lives of learners during the pandemic. Our finding that medical students were more likely to use Social Critique processes has multiple parallels in the literature. If leaders in academic medicine desire to create enlightened change agents through transformative learning, such education must continue throughout graduate medical education and beyond.

## Introduction

The coronavirus disease 2019 (COVID-19) pandemic is having a profound impact on the personal and professional lives of individuals around the world. Particularly affected are those in the healthcare field [1]. However, effects have varied by level of training. Physicians have been called to work longer hours [2], utilize telehealth platforms [3,4], and/or be placed in high-risk clinical environments [5]. On the other hand, medical students were removed from direct clinical patient experiences altogether, often finding new roles within the community [6]. The perspectives of students and other trainees early on during the pandemic have been captured in the medical literature [7]. However, despite the significant amount of publications on COVID-19 in the past year [8], there remains a paucity of empirical quantitative and theoretically-grounded data on the transformative effects of the pandemic on frontline healthcare workers, especially those in training.

Transformative Learning (TL) theory explains the process by which individuals experience "dramatic, fundamental changes in the way we see ourselves and the world in which we live." [9, p166]. This may manifest in an individual ultimately acting differently, having a deeper sense of self-awareness, developing more open perspectives, or experiencing a deep shift in worldview [10] According to the theory, this process is set in motion by a disorienting dilemma. While disorienting dilemmas are usually individual in nature and unique to the learner (e.g. the death of a family member, being a victim of a crime) [11], COVID-19 is distinctive in that it potentially serves as a single disorienting dilemma for a great number of people. Multiple articles have already reported the psychological impact of COVID-19 on healthcare workers [12–14]. Thus, COVID-19 offers a rare opportunity to better understand how different groups experience and respond to the same life-changing event in the workplace and classroom. Such understanding will help physician leaders inform practical changes that may have positive downstream effects on employees and learners alike, in the setting of this pandemic as well as other future disruptive scenarios. We believe that understanding where and how transformative learning occurs for students and housestaff within the clinical, educational, and home environments will provide these data.

### Background

Transformative learning is traditionally thought of as occurring through one or more of three processes: Cognitive Rational, Extrarational, and Social Critique [10,11,15] The cognitive rational process is based on the learner experiencing a disorienting dilemma, critically reflecting on their assumptions that created the dilemma, and engaging in reflective discourse with others, followed by some sort of action. The extrarational process involves bringing consciousness to the previously unconscious through inner psychic exploration resulting in insights or epiphanies, e.g., through art, storytelling, dreams, and spiritual endeavors. The social critique

process views TL as empowerment to challenge oppression. Each of these three processes of TL can be further differentiated into more specific subdomains (S1 Table).

## Aims of the study

The primary aim of our study was to examine the relative types of transformative learning outcomes and processes experienced by learners at an academic medical center during COVID-19. An additional aim was to determine if there was a difference between students and housestaff in the degree to which specific TL processes and outcomes were experienced or achieved respectively. Finally, we aimed to use qualitative analysis to explore the features underlying the reasons for the above.

## Methods

### Study design, participants and procedure

We used a parallel convergent mixed-methods study design to perform a cross-sectional survey of physician assistant (PA) students, medical students, residents, and fellows at an academic health center that serves a geographically diverse population (urban to rural). This study design involved collecting both quantitative and qualitative responses [16]. We collected data in September and October, 2020. The pool from which our sample was drawn was made up of 63 PA students, 615 medical students, and 661 residents and fellows. At our institution of study, the 2019 entering class of PA students had an average age of 26, were 60% women, 47% were classified as underrepresented or disadvantaged students, and 17% were Pennsylvania residents [17]. The 2019 entering class of medical students were 61% women, 7% underrepresented minorities, 2% had a military background, and 37% were Pennsylvania residents (age data not available) [18]. Both PA and medical students have similar pre-requisites, and training structure in both programs includes a mix of classroom and clinical activities. Moreover, COVID-19-related clinical care restrictions at our institution during the period of study were similar between these groups. Given these similarities, and to increase statistical power, medical and PA students' responses were grouped together for analysis (hereon referred to as 'students'). While much of student learning was shifted to online and virtual platforms as they were removed from direct clinical care during the first peak of the pandemic, residents and fellows continued clinical work, with some providing direct and indirect care of COVID-19 positive patients. For the reason, graduate medical learner responses from residents and fellows were also grouped together (hereon referred to as 'housestaff'). Recruiting materials were distributed via respective listserv emails. Of note, due to the large volume of recruitment emails sent to learners for COVID-19 educational research, our institution implemented a policy limiting the amount of recruitment emails to students to one initial email with two weekly reminder emails. This policy was enacted prior to our recruitment period, and for standardization, we held to this construct for housestaff recruitment as well. As this was a pilot study intended to inform our future work studying transformative learning during COVID-19, formal power calculations were not performed, and we did not set a goal sample size.

### Ethics

The Penn State College of Medicine Human Subjects Protection Office determined that the study (STUDY00015312) met the criteria for exempt research. Due to the potential for significant emotional responses by participating in research related to COVID-19, contact information for local mental health resources was provided on the consent page.

## Measurements

We used the Transformative Learning Survey (*transformativelearningsurvey.com*, TLS) [19] a mixed-methods data collection tool intended to assess the outcomes and processes related to transformative learning experiences. TLS was developed by HS and colleagues under a prior protocol (CATS IRB #00036934); measurement validity and reliability have been reported previously [10].

TLS assesses processes and outcomes of transformative learning [10]. TLS assesses the outcome of transformative learning across four subdomains. Such transformation may take the form of a respondent developing increased self-awareness, acting differently, adopting increased openness, or undergoing a shift in worldview. Additionally, TLS breaks down the process by which respondents experience TL according to three domains corresponding to the traditional TL processes discussed above. These outcomes and processes are assessed via a randomly ordered 90-item questionnaire utilizing a 4-point Likert-type scale, as well as two open-ended questions that we modified to fit the purpose of this study. The survey concludes with a short demographics section. After survey completion, participants receive a graphical analysis of their survey results.

## Data analysis

**Quantitative data.** Likert-type scale scores were converted to a 0–100 point scale for analysis. We used independent-sample t-tests for equality of means to assess group differences in the use of processes and achievement of the outcome of transformative learning. We first assessed group differences in TL processes and outcome. If a significant difference was found in a process or the outcome of TL between groups, we then used the same statistical tests to examine differences at the subdomain level within that process or outcome.

**Qualitative data.** We performed a qualitative content analysis of responses to the two open-ended questions: (1) "In what ways has COVID-19 altered your life in a meaningful way?" (2)" Describe this experience with COVID-19. Was/how was it life changing? Who was involved? In what context did this change occur?". A codebook was developed by BS based upon a reading of 100% of the open-ended responses and collaboratively reviewed and edited by all co-authors until agreement was reached on a final codebook. BV and BS then individually coded all open-ended response data. Our initial coding achieved modest interrater reliability for students ($\kappa = 0.55$) and housestaff ($\kappa = 0.59$). Disagreements were resolved by selecting various occurrences of low agreement and further discussing discordantly coded responses until a final code was assigned. Final adjudications were made by the entire study team. This process was repeated until a Cohen's kappa statistic for inter-rater reliability ($\kappa$) of greater than 0.7 was achieved [20]. After coding calibration, we ultimately achieved our goal for inter-rater reliability (students, $\kappa = 0.73$; housestaff, $\kappa = 0.79$). NVivo 12 and Release 1.3 (QSR International, Australia) were used to support the analysis of qualitative data.

Chi-square tests were performed to analyze differences in the number of quotes that fell into the major coding categories, and differences between students and housestaff in the proportions of responses that fell into the major coding categories. The Statistical Package for the Social Sciences version 26.0 (IBM, Armonk, NY, USA) was used for all statistical tests, with the level of significance set to $p < 0.05$.

# Results

## Demographic characteristics

The pool from which our sample was drawn was made up of 63 PA students, 615 medical students, and 661 residents and fellows. One participant's data in the housestaff group was not

captured correctly by the survey website resulting in confusion as to which answers corresponded to which questions, and thus was deleted. Another participant within the student group did not provide demographic data, but was retained in the analysis of scale scores. Twenty-two students (3.2%) and eleven housestaff (1.7%) completed the survey, for a total of 33 participants included in our analysis (Table 1). Overall, the sample was mostly White, and from the US. Housestaff were older, with more education and occupational attainment.

## Quantitative results for transformative learning

**Overall results.** Across all learners studied, there were significant ($p < 0.05$) differences among TL outcomes; most prominent was Self-Awareness, Acting Differently was intermediate, and Openness and Worldview Shifts were lowest (Table 2). There also were significant ($p < 0.05$) differences among TL processes experienced by our whole sample; Cognitive Rational and Social Critique processes were more prominent than Extrarational processes (Table 2). S2 Table shows that the six most prominent process scales were evenly divided between the Cognitive Rational and Social Critique domains (with Discourse significantly higher than all other processes), while three of the Extrarational processes were among the four lowest ranked.

**Subgroup comparisons.** There were no statistically significant differences between the student and housestaff *outcome* measures of transformative learning (Table 3). No significant differences were observed between student and housestaff use of the Cognitive Rational or Extrarational processes of TL. However, students in our sample were more likely than housestaff to have undergone transformative learning due to COVID-19 through the *process* of

**Table 1. Baseline characteristics of study participants.**

| Characteristic | Students (N = 21) | Housestaff (N = 11)* |
|---|---|---|
| Sex–n (%) | | |
| Female | 14 (66.7) | 5 (45.5) |
| Male | 7 (33.3) | 6 (54.5) |
| Age category–n (%) | | |
| 18–24 | 7 (33.3) | 0 (0.0) |
| 25–34 | 12 (57.1) | 8 (72.7) |
| 35–44 | 2 (9.5) | 3 (27.3) |
| Highest level of education–n (%) | | |
| Bachelor's degree or less | 15 (76.2) | 0 (0.0) |
| Graduate degree | 5 (23.8) | 11 (100.0) |
| Race‡ ¯n (%) | | |
| White | 12 (57.1) | 8 (72.7) |
| Nonwhite or N/A | 9 (42.9) | 3 (27.3) |
| Nationality† –n (%) | | |
| United States | 21 (100.0) | 8 (80.0) |
| Other | 0 (0.0) | 2 (20.0) |
| Employment–n (%) | | |
| Professional | 1 (4.8) | 11 (100.0) |
| Student | 20 (95.2) | 0 (0.0) |

* N = 10 for housestaff respondents to nationality.

‡ Available options included White, Black, Latino/Spanish, Asian / Pacific Islands, First Nations / Native American / Aboriginal, Mixed Race, and Other. Responses to Other included Ashkenazi Jew and N/A. Data on race are reported as such for statistical purposes and brevity given the demographics of our health center and are not intended to imply White as normative.

† Available options included United States and Other (with a free text option).

**Table 2. Ranked means for TLS outcome scales and process domains.**

| TL Outcomes | Mean | SD | groupings |
|---|---|---|---|
| Self-Awareness | 59.80 | 24.9 | a |
| Acting Differently | 55.56 | 25.3 | a, b |
| Openness | 52.32 | 20.3 | b |
| Shift Worldview | 48.08 | 27.3 | b |
| **TL Processes** | **Mean** | **SD** | **groupings** |
| Cognitive Rational | 68.12 | 13.5 | c |
| Social Critique | 64.90 | 16.3 | c |
| Extrarational | 53.45 | 16.8 | d |

Table 2 depicts the mean TLS scores showing which TL outcomes were most achieved compared to others, as well as which TL processes were most frequently utilized across our entire sample.

NOTE: All measures are grouped so that line items can be compared with other line items within the same heading. All measures with the same grouping letter are not significantly different from each other (p > 0.05). Measures not sharing the same grouping letter are significantly different from each other (p ≤ 0.05, two-tailed).

Social Critique (M ± SD: vs 69.32 ± 16.1 vs 56.06 ± 13.1; p = 0.025). Analysis of the subscales within this domain, there were no differences between groups for Empowerment or Unveiling Oppression, but students were more likely than housestaff to have gone through the sub- processes of Social Action (77.0 ± 15.40 vs 62.4 ± 17.45; p = 0.023) and Ideology Critique (75.0 ± 18.34 vs 57.6 ± 16.94; p = 0.010).

## Qualitative results for transformative learning

To better understand the content driving the quantitative assessment of TL outcomes and processes above, the following section details our qualitative and mixed methods analyses of the two open-ended survey questions.

**Overall results.** The open-ended question responses revealed four major content areas as defined in our inductive coding scheme (Table 4). These responses included negative change (67.7%), positive change (18.0%), existential change (8.4%), and no change (6.0%); responses sum to over 100% due to rounding error. Negative changes were significantly more frequent than positive, existential and no change (p < 0.001), and positive changes were significantly more frequent than existential and no change (p < 0.01). We also examined whether students and housestaff differed in the proportions of their responses in various categories and codes/subcodes; the only significant difference was between the proportion of student (74.8%) and housestaff (53.6%) comments that were negative (p < 0.01). In the following sections, we analyze the open-ended responses according to the four main content areas, using exemplary quotes where applicable.

**Table 3. Tests of group differences in TL domain means.**

| | Student | | Housestaff | | |
|---|---|---|---|---|---|
| Measure | Mean | SD | Mean | SD | p-value |
| Outcomes | 52.12 | 20.8 | 54.85 | 21.3 | .729 |
| Cognitive Rational Processes | 66.67 | 12.1 | 68.85 | 14.4 | .669 |
| Extrarational Processes | 50.30 | 17.7 | 55.03 | 16.5 | .455 |
| Social Critique Processes | 69.32 | 16.1 | 56.06 | 13.1 | .025 |

**Table 4. Qualitative data map.**

| Codes and Sub-codes | Definition | Exemplary Quote | Total Refs | Student Refs | HS Refs |
|---|---|---|---|---|---|
| **All Negative Comments** | | | 113 (67.7%) | 83 (74.8%) | 30 (53.6%) |
| All education stressors | | | 39 | 30 | 9 |
| Effects on learning experiences | References to shift to virtual learning; less opportunities for learning experiences; changes in patient interaction (distancing, telemedicine, masks and communication, etc.) | "I am seeing my clinical year patients via telehealth for some services" [S] | 20 | 14 | 6 |
| Effects on non-experiential learning and career | References to broad mentions of changes in their program, uncertainty, worries of this change impacting their growth to next step in their career, changes in grading system, effects on their studying timeline for important milestone exams | "Not being able to graduate with my medical school classmates in person was a bummer because it was a literally a day I had envisioned for a solid decade" [HS] | 11 | 9 | 2 |
| Other educational stressors | Any educational stressor mentioned not included in codes above. | "It has also greatly increased stress associated with my education" [S] | 8 | 7 | 1 |
| All social isolation/ lack of connection | | | 32 | 21 | 11 |
| Family/loved ones | References to lack of connection with family/loved ones; Can't travel to see loved ones; can't be around them for safety purposes; worry for their physical safety | "less experiences with family in traveling or even seeing my family members who are high risk" [HS] | 19 | 11 | 8 |
| Decline in mental health, depression, isolation | References to less social interaction, change in how they interact with others, feeling lonely and/or isolated, making less friends due to restrictions, any general references to mental health. | "which has affected mental health. I find that I have more social anxiety being out in public and in crowds than previously" [S] | 5 | 4 | 1 |
| Other social isolation/lack of connection | Anything negative mentioned not included in codes above. | "COVID has made me feel distant from everyone regardless of how close we may be" [HS] | 8 | 6 | 2 |
| Changes in daily life | References to changes in how someone does shopping for necessities, doctor appointments, attending church, lack of extracurriculars, not going to the gym/ less active, not going to restaurants, not pursuing hobbies. NOT related to education changes. | "COVID has made me stop shaking hands" [HS] | 29 | 23 | 6 |
| Financial stressors | References to job loss, unemployment, loss of income/ benefits, etc. | "Additionally, my significant other is not a medical student like myself and relies on their employment for ensuring bills, rent, etc. are paid for, so financially it has been quite scary for both of us" [S] | 6 | 5 | 1 |
| Other negative comments | Anything negative mentioned not included in codes above. | "more TV time than I would like for myself and young child, increased weight gain due to less activity" [HS] | 7 | 4 | 3 |
| **All Positive Comments** | | | 30 (18.0%) | 18 (16.2%) | 12 (21.4%) |
| Opportunity to spend more time with family | References to more time to spend with family members, both in-person and virtually. | "COVID 19 has brought some positive change, however. I got to spend more time with my family and really bond with them as a result of the last few weeks of didactic year being primarily online" [S] | 10 | 6 | 4 |
| Opportunity for improved work-life balance | References to shift to virtual landscape being positive; being able to multi-task due to virtual school/ meetings/work; virtual space making someone more efficient with their tasks. | "Work from home has increased productivity" [HS] | 2 | 1 | 1 |
| Opportunity for investment in myself | References to having time 'for themselves,' opportunity to slow-down, spending more time outdoors, more active. | "improving various aspects which provide me with more time to do things that I WANT to do" [S] | 6 | 3 | 3 |
| Other positive comments | Anything positive mentioned not included in codes above. | "Better to environment" [HS] | 12 | 8 | 4 |
| **All Existential Comments** | | | 14 (8.4%) | 7 (6.3%) | 7 (12.5%) |

(*Continued*)

**Table 4.** (Continued)

| Codes and Sub-codes | Definition | Exemplary Quote | Total Refs | Student Refs | HS Refs |
|---|---|---|---|---|---|
| Awareness of humanity | References related to the humanity of the pandemic situation, of the suffering, of one's own humanity/ existence, 'putting things into perspective.' Related to humanity needs to 'wake up' and make changes. | "Every year as an adult has granted me less faith in policy making and administration" [HS] | 7 | 2 | 5 |
| Broad scale global changes | References to large scale changes in a global level or societal level; farther reaching changes larger than themselves. References to a 'new normal.' | "I think it will take a few more pandemics and major natural disasters before people get to that point" [HS] | 1 | 0 | 1 |
| Structural racism and violence | References to the pandemic making them more aware of structural racism, structural violence, marginalization, etc. | "My eyes have been opened to social injustices and structural racism" [S] | 2 | 2 | 0 |
| Other existential comments | Any existential comments not included in codes above. | "It made me appreciate the life we had and the freedom of traveling that we had and not socially distancing" [HS] | 4 | 3 | 1 |
| **All Neutral/Null Comments** | | | 10 (6.0%) | 3 (2.7%) | 7 (12.5%) |
| Neutral/Null Comments | No change or impact, neutral outlook | "once I returned to rotations in June 2020, it was mostly normal" [S] | 10 | 3 | 7 |
| **TOTAL CODES** | | | 167 (100%) | 111 (100%) | 56 (100%) |

NOTE: Dark shaded cells represent aggregation of codes within the major categories. Light shaded cells represent codes. Unshaded cells represent subcodes, which are aggregated into codes.

**Negative change, shift, or perception.** Our population of medical trainees experienced many challenges related to the COVID-19 pandemic. The most common negative changes were educational stressors (23.4%), social isolation (19.2%), and changes in daily life (17.6%).

Educational stressors were cited as life changing. More commonly, these pertained directly to learning experiences. Notably, some housestaff directly cared for COVID-19 positive patients, describing them as the sickest patients they had ever seen, or participating in the most difficult end-of-life discussions they have ever had. Staying up to date on evidence based medicine relating to care of COVID-19 positive patients also proved challenging.

*I was in [direct] contact since March/April 2020 and treated them almost exclusively for a few months. It was the most unique experience I've had in medicine since I started as a student and resident. The patients were the most sick that I had ever seen. Knowledge was constantly changing and very significant attempts to stay even more up to date than usual were difficult. Also, the hospital environment was constantly "on edge" and people were scared often. The force of individuals trying to help take care of people however was tremendous and valiant at minimum. They were the hardest patients to treat in the ICU.* [Housestaff (HS)]

Relationship-building with patients was also difficult. Telehealth was cited as a challenge, though increased skills in this realm was occasionally seen as a positive. Another participant felt that shift to synchronous online learning was discordant with her or his preference for group learning. Students frequently commented on shorter learning experiences or decreased direct clinical experiences.

Negative learning experiences were not limited to the classroom. Trainees were not naïve to the public relations and leadership challenges faced by hospital administration, and were disheartened by some of the pandemic responses they observed.

*However, at first many of us were very surprised that the hospitals were so hesitant to offer information to the public for what appeared to be worry about PR issues (unfortunate). In fact many physicians and medical staff were very frustrated with our institutions for their lack of organization and progressive perspective initially. Things did improve over time but it was clear that most hospital systems were not prepared to accommodate these rapid changes. This also occurred in the research world when we got up and running with plasma trial, etc. Our hospital has many years ago gotten rid of [its] "pandemic team" which would have likely changed a lot of our initial responses.* [HS]

Students and housestaff alike also described structural changes to their education and resultant career implications related to the pandemic. Schedule and curricular changes, alteration in grading structure, licensing examination delays, shifts to virtual residency interviews, and cancellation of in-person graduation all significantly affected our trainees.

*My Step 1 exam was delayed multiple times—I was supposed to take it in March, but it was cancelled repeatedly and I ended up taking it at the end of July. The ongoing delay—and perpetual studying—was stressful, but I was still surprisingly able to do well on the exam. I was supposed to take Step 2 in June and now it's in November. Also, a significant change is the switch to virtual interviews this fall/winter for residency applications. While I'm grateful for the savings, I'm mostly disappointed because we will miss out on the opportunity to visit programs in-person and meet residents/faculty at those programs. I worry it'll be challenging to find the program that is right for me, and it's also unnerving in general to be part of such an unprecedented application cycle.* [S]

As expected, challenges to mental health were also a concern for our participants, though this was commented on far more frequently by students than housestaff. This manifested as impaired quality of life, loneliness, social anxiety, depression, and an inability to cope with stress.

*I find that I have more social anxiety being out in public and in crowds than previously.* [S]

Many of these mental health concerns seemed to stem from social isolation from family members or other loved ones. Trainees commented that they ensured proper social isolation for fear of harming vulnerable or immunocompromised individuals, both young and old. Some resorted to video calls or other electronic means of communicating. However, these were not felt to be an adequate substitute for in-person interaction, especially for communicating with older relatives who may not be facile with technology.

*I miss seeing my grandparents. I'd love to visit them, but I'm afraid I'll accidentally give them COVID. It's just hard to talk to them because they don't have the technology to video chat and their poor hearing makes it hard to talk on phones.* [S]

Responses coded as changes in daily life were also commonly referenced by our trainee population as altering their life in a meaningful way. Examples included challenging interpersonal communication due to facial coverings (masks), decreased physical contact (e.g. shaking hands), prioritizing only essential commerce, cancellation of major life events (e.g. vacations, honeymoon), and increases in screen time. Some cited these changes leading to a decrease in physical health manifested by weight gain. Significant life changes for family members of trainees were also a cause of trainee stress. Not being able to attend children's medical

appointments, and challenges related to children shifting to remote learning were also cited as meaningful life alterations. Another frequently referenced challenge, especially during such a stressful event, was the removal of many coping mechanisms for stress such as public gyms or group sports, dining out, concert attendance, and organized religion.

> *I have been unable to go to church, which is something I have been doing weekly for my entire life. To see how abruptly the religious institutions were mandated to close, and how other (much less critical to me) places like animal grooming centers and liquor stores were deemed essential and remained open was a little shocking. As time went on movie theaters and restaurants and hair salons opened, and the churches remained prohibited to open. Visitors were allowed to re-enter healthcare settings in a limited way, but visiting church members for moral support was still discouraged by local government leaders. And of course, the local and worldwide church leaders obeyed.* [Student (S)]

Perceived and actual career challenges were also noted with one student worrying that "things may be affected for our job hunt that is coming up soon," with another noting, "I'm less likely to want to meet with potential mentors over Zoom after having classes online all day."

Our participants also commented on the financial impact of COVID-19. Examples of this included personally losing money, food insecurity, or fear of pandemic-related job loss for a non-medical partner.

> *It was life changing economically mostly because having money for food and housing is an immediate concern. The health risks almost take a back seat when you have to worry about losing a source of income, health insurance, etc.* [S]

**Positive change, shift, or perception.** Despite the drastic change to daily life that many felt due to COVID-19, our data showed that students and housestaff also recognized several areas of positive change in their lives, although each was 6.0% or less of coded responses. The increased reliance on technology in multiple aspects of our participants' lives did demonstrate potential for improved work-life balance.

> *Better work(school)/life balance. Learned the value of online shopping and online grocery ordering/pick-up.* [S]

The 'pause' related to the pandemic also afforded our participants the opportunity to invest in themselves, discover new hobbies, and ultimately do more of what they wanted to do. Many commented on new ways to keep entertained while businesses were shut down. These included fitness, cleaning, home repair, and reading. One student even commented on an increased involvement in self-care.

> *I think the more positive aspect was some of the time I got to [spend] to myself. I never really had the opportunity to sit and enjoy being by myself, and doing things for me. It gave me more time for self care.* [S]

Despite social isolation from some family members, the opportunity to spend time with other family members increased drastically. This was described as increased in-person time with a spouse at home due to quarantining, or connecting with distant friends and family

through electronic means. Both our populations recognized themselves growing closer with their family, and specifically mentioned increased time to focus on their closely held priorities. Increased outdoor activity with family members also seemed to be an avenue for increased wellness.

*I have spent more time focused on faith and family, which are ultimately my life's greatest priorities.* [S]

**Existential change, shift, or perception.**  In addition to the modest positive and negative changes to our study participants' lives, COVID-19 led to a large-scale shift in some participants' beliefs about the world that were indicative of transformative learning; 8.4% of coded responses fell into this category. This is the only content area that we interpreted as transformative learning. This included developing a greater awareness of humanity, recognition of broad scale global changes, or an appreciation of structural racism and violence. Responses representing awareness of humanity included reflections on the influence of social pressures on medical experts, institutions, and policy-makers, ultimately leading to a decreased faith in the system.

*Showed me how leading medical experts and institutions have little to no interest in operating on [an] evidence basis and adapting as more information is acquired, and instead are more in favor to conforming to an underlying social moral.* [HS]

Some of these comments recognized classmates' not obeying public health regulations ("We were required to quarantine for two weeks upon arrival, but many people in my class did not"). A more powerful statement was that "It made me very aware of the value of life and the meaning of mine." One student was able to appreciate how, while these are unprecedented times for most living today, "Humans have dealt with plagues since we developed cities."
A few comments focused on how the pandemic was life-changing on a broader, more global scale.

*Absolutely life changing. It is something that we hope only really happens once every few generations. The world shut down, then changed, then returned to a 'new normal' all within the span of six months or so.* [S]

Another comment indicated that "it will take a few more pandemics and major natural disasters" for society at-large to recognize the issues of sustainability with our current way of life.
Finally, some comments indicated that the COVID-19 pandemic opened their eyes to issues of racial inequities and social justice and inspired them to act.

*COVID-19 opened the discussion for structural racism in the healthcare setting as well as in other settings. I became more aware of what is going on and more vocal about what we can do to change it.* [S]

**No change, no impact, neutral outlook.**  Interestingly, despite the large scale structural or logistical changes from COVID-19, some comments (6.0% of all comments) reported that the pandemic had not altered their life in a meaningful way.

*I haven't had direct experiences with Covid-19 and am fortunate that none of my loved ones have been ill. In many ways, my life has stayed the same except for some logistical changes.* [S]

Notably, one student felt she or he "might later say that my life was changed by COVID but for now, it's just another thing to deal with."

## Discussion

Our mixed-methods study evaluating the pandemic-driven transformative learning of our institution's students and housestaff had multiple interesting findings. Both our aggregate data and group differences provide valuable information for medical educators.

With regard to our aggregate quantitative results, the outcome subdomain of TL that our population most frequently achieved was that of greater Self-Awareness (Table 2). This may be explained by the vast opportunities for introspection afforded by quarantining and social distancing. With regard to the processes of TL, survey participants most commonly underwent transformative learning through the Cognitive Rational and Social Critique domains, and thus, the Extrarational process was least utilized (Table 2). This is not surprising, given that the latter process often involves making meaning of experiences through the arts, imagination (such as dreams), or spiritual endeavors. Activities that may have supported revelations through this process were frequently deemed non-essential during the early part of the pandemic. This is supported by comments from our participants noting the impact had by cancellation of organized religious services on emotional wellbeing.

Qualitative data coded to negative changes or perceptions far exceeded that which we coded to positive, existential, or no change. This finding is interesting, given that transformative learning theory traditionally focuses on positive transformation leading to a more open and inclusive worldview [9]. However, some argue that after an initial disorienting dilemma, a slower transformative process may take place for some individuals, requiring additional catalysts for new worldviews to be adopted [21]. At the time we are writing this manuscript, we are still in the midst of COVID-19. Thus, we suspect not enough time may have elapsed for some individuals to truly make meaning of their experiences or experience transformation. We hypothesize that if this study were repeated, there may be an increase in transformative changes and a difference in Extrarational process. Additionally, some of our survey responses indicated that our participants may be in their earlier years of medical or PA school. While graduate medical curricula explicitly [22] or implicitly [23,24] grounded in transformative learning theory have the ability to reconnect residents to the values that drew them to medicine or their particular specialty, students may still be in a formative rather than transformative stage of their professional identity formation [25,26]. Given that the majority of our participants were students, this may have shifted our results away from the traditional assumptions of TL theory.

Within the category of negative change, qualitative data coded as educational stressors were most common. This does have some parallels in the existing literature. One multicenter German study of medical students found that the majority of those surveyed reported significantly more pandemic-related distress pertaining to their studies than their private lives [27]. Specifically, students in their study were more distressed about potentially missing out on study materials or losing a semester due to COVID-19 than they were on missing aspects of personal lives. Moreover, our results showed that significantly more students had open-ended question responses coded as negative change. This may be due to the fact that undergraduate medical trainees experienced far more training disruption due to accrediting society protective mandates than those in graduate medical education.

Further research should be undertaken evaluating similarities and differences in results when using the Transformative Learning Survey to evaluate negative experiences in healthcare with that of the Moral Injury Symptom Scale-Healthcare Professionals (MISS-HP) [28], a tool designed to evaluate such negative experiences. This may provide insight as to whether moral injury is a form of negative transformative learning.

## Differences between students and housestaff

Quantitative analysis indicated that students were more likely than residents and fellows to make meaning of life-changing events through the TL process of social critique. There are multiple potential reasons for this. The attention of undergraduate medical students to social justice is well reported in the literature [29,30], and curricula on this topic may even be student-driven [31]. Notably, *White Coats for Black Lives* is a national *medical student* organization [32], and concepts such as socially-accountable health professional education [33,34] and health systems science [35–38] seem to be more closely associated with undergraduate medical education. Even during the current pandemic when students were removed from clinical activities at the direction of accrediting bodies, many assumed value-added health system roles, including those that directly contributed to the social welfare of the community [39]. Additionally, residents whose roles are traditionally more clinically-oriented than medical students may be more likely to be exposed to unprofessional conduct through the hidden curriculum [40]. This may lead to an increase in cynicism in graduate medical trainees [41], which we hypothesize may detract from trainees' optimism in realms such as social change. Alternatively, residents and fellows may have been busier than students and simply did not have enough opportunity for critical reflection.

## Implications for practice

The utilization of a mixed methods approach allowed us to identify that the majority of negative changes reported in our open-ended questioning surrounded educational changes due to COVID-19. While some of this may have been unavoidable, it is up to academic health center leadership to provide clear communication during crises (while balancing the avoidance of information overload) and implementing strategies to minimize the educational impact on its trainees.

Additionally, while many academic health centers such as ours have been quick to recognize the mental health implications of the pandemic and provide support for their frontline workers, there appears to be a lack of evidence supporting the selection of beneficial interventions [42]. Moreover, prioritizing resilience for mental health may be more focused on the treatment of a symptom, rather than addressing the underlying cause. While academic health center leadership have only limited control over the spread of COVID-19 and other diseases, it is well within their purview to provide support for their learners by way of replacing what was lost due to pandemic-related restrictions. Our population of trainees frequently referenced lack of opportunities for group fitness activities or the ability to congregate to practice their faith. While recognizing the congregation limits set by public health guidelines, we believe these to be an underdeveloped means of support for trainees during extreme times of stress. After recruitment for our study was completed, our Department of Medicine implemented an incentivized team-based fitness tracking program until in-person fitness restrictions are lifted, at which time the department will pilot free on-campus fitness center memberships for the department. We believe similar innovative programs could hold promise at other institutions as well. While our institution has a robust network of affinity groups and a successful diversity grand rounds series that have continued to operate virtually during the pandemic, there have

been some suggestions that an adequate replacement for medical trainees whose practice of their faith has been interrupted by the current crisis may be lacking. Perhaps tapping into resources within our clinical pastoral care training programs may lend well to innovative ideas that would benefit both students and housestaff alike.

In our qualitative analysis, we revealed that multiple trainees across both populations surveyed reported that COVID-19 was not life-changing in any meaningful way. Moreover, with the exception of responses coded to the existential change category, we felt that most of our participants' responses did not signify transformative learning experiences according to theoretical constructs. Previous literature on TL within health professions education have reported individuals who showed no evidence of transformative learning [43], or who either underwent cognitive or behavioral change without transformation or seemed unaffected by their disorienting dilemmas [44]. Struggle with uncertainty and challenges of ethical issues may play a role in who has a transformative experience and who does not [43]. In our study, we posit that enough time may not have elapsed since the disorienting dilemma to truly allow for adequate critical reflection since the pandemic was ongoing at time of survey completion. Similarly, we believe adequate space for critical reflection may not have been created. Discourse, an integral component of Cognitive Rational TL theory, was the most prominent component of the TL process for our sample (S1 Table). Thus, perhaps those without an educator figure or guide to assist in the TL journey may find their process stalled. Further research is needed in this area.

## Strengths, limitations, and contribution to the field of mixed methods research

Our study provides both quantitative and qualitative data as it relates to undergraduate and graduate medical trainee transformative learning during a large-scale global crisis. Not since September 11th has a single historical event served as a disorienting dilemma for such a great number of people [45]. Our study adds to such literature using a valid and reliable survey instrument. Most commonly, literature on transformative learning is self-reported [9]. While we were able to gain valuable insight on the content of transformative experiences through our qualitative analysis of open-ended questions, our study importantly adds rare quantitative data pertaining to TL to the medical literature.

Our study had multiple limitations. As with any voluntary survey, our study was prone to selection and response biases. Moreover, our small sample size limited our power to detect significant between-group differences in TL measures. While we grouped PA and medical students' responses together due to perceived group similarities, public admissions data indicates there may have been a far greater percentage of underrepresented minorities in our pool of PA students, though gross numbers were similar (14 underrepresented PA students versus 11 underrepresented medical students). Nevertheless, this impacts the ability to draw conclusions on the impact of COVID-19 on underrepresented students. We did not collect the breakdown of participants other than the standard demographic items included in the survey, and we do not have the same data on the nonrespondents. Thus, we are unable to empirically address the issue of representativeness. While our study took place in a large academic health center, it had a relatively low volume of COVID-19 positive patients early on in the pandemic [46]. Thus, this limits our external validity, especially for our quantitative results. However, there is continued debate as to whether and how qualitative results can or should be generalized [47]. While our recruitment was limited by institutional restrictions on survey emails to trainees implemented after COVID-19, we achieved qualitative saturation in our findings based on the number of references for each sub-code and the minimal amount of uncoded data in our content analysis. Finally, while our open-ended questions provided valuable qualitative data, terse responses and the inability to ask follow up or clarifying questions limited our data collection.

## Conclusion

Through the lens of transformative learning theory, our study provides valuable insight into the personal and professional lives of PA students, medical students, residents, and fellows from our large academic medical center during COVID-19. While students may be more likely overall than housestaff to undergo transformative learning through the process of social critique, housestaff also reported substantial transformative learning on most measures. Meanwhile, all trainees struggled with challenges both to their educational experiences and their daily lives. These challenges related to COVID-19's individual transformative effects shed light on various entry points where academic health center leadership can intervene with the hope of creating resilient learning communities and workplaces throughout this pandemic and future challenges that will inevitably arise.

## Supporting information

**S1 Table. Definition of transformative learning concepts [10].** This table provides definitions for the various outcomes and processes of Transformative Learning Theory.
(DOCX)

**S2 Table. Ranked means for TLS processes.** NOTE: All measures with the same grouping letter are not significantly different from each other ($p > 0.05$). Measures not sharing the same grouping letter are significantly different from each other ($p \leq 0.05$, two-tailed). C, Cognitive Rational process domain; S, Social Critique process domain; E, Extrarational process domain.
(DOCX)

**S1 File.**
(PDF)

## Author Contributions

**Conceptualization:** Benjamin Vipler, Bethany Snyder, Jennifer McCall-Hosenfeld, Paul Haidet, Heather Stuckey.

**Data curation:** Mark Peyrot, Heather Stuckey.

**Formal analysis:** Benjamin Vipler, Bethany Snyder, Mark Peyrot, Heather Stuckey.

**Funding acquisition:** Benjamin Vipler, Jennifer McCall-Hosenfeld, Paul Haidet, Heather Stuckey.

**Investigation:** Benjamin Vipler.

**Methodology:** Benjamin Vipler, Jennifer McCall-Hosenfeld, Paul Haidet, Heather Stuckey.

**Project administration:** Benjamin Vipler, Bethany Snyder.

**Software:** Benjamin Vipler, Bethany Snyder, Mark Peyrot.

**Supervision:** Jennifer McCall-Hosenfeld, Paul Haidet, Heather Stuckey.

**Visualization:** Benjamin Vipler.

**Writing – original draft:** Benjamin Vipler.

**Writing – review & editing:** Benjamin Vipler, Bethany Snyder, Jennifer McCall-Hosenfeld, Paul Haidet, Mark Peyrot, Heather Stuckey.

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
