## [Decision Letter · Decision Letter 0]

21 Mar 2022

PONE-D-21-19528Transformative Learning of Medical Trainees during the COVID-19 Pandemic: A cross sectional surveyPLOS ONE

Dear%,

Thank you for submitting your manuscript to PLOS ONE. After careful consideration, we feel that it has merit but does not fully meet PLOS ONE’s publication criteria as it currently stands. Therefore, we invite you to submit a revised version of the manuscript that addresses the points raised during the review process.

 Please submit your revised manuscript by May 5, 2022. If you will need more time than this to complete your revisions, please reply to this message or contact the journal office at plosone@plos.org. Please include the following items when submitting your revised manuscript:A rebuttal letter that responds to each point raised by the academic editor and reviewer(s). You should upload this letter as a separate file labeled 'Response to Reviewers'.A marked-up copy of your manuscript that highlights changes made to the original version. You should upload this as a separate file labeled 'Revised Manuscript with Track Changes'.An unmarked version of your revised paper without tracked changes. You should upload this as a separate file labeled 'Manuscript'.

We look forward to receiving your revised manuscript.

Kind regards,

Muhammad Shahzad Aslam, Ph.D.,M.Phil., Pharm-D

Academic Editor

PLOS ONE

Journal Requirements:

2. Thank you for stating the following financial disclosure: "Funding for this study was awarded (BV) through a Penn State Department of Medicine (https://med.psu.edu/medicine) Inspiration Pilot Grant for COVID-19 Related Research (INSPIREVIP2020).  The funders had no role in study design, data collection and analysis, decision to publish, or preparation of the manuscript."

We note that one or more of the authors is affiliated with the funding organization, indicating the funder may have had some role in the design, data collection, analysis or preparation of your manuscript for publication; in other words, the funder played an indirect role through the participation of the co-authors. If the funding organization did not play a role in the study design, data collection and analysis, decision to publish, or preparation of the manuscript and only provided financial support in the form of authors' salaries and/or research materials, please do the following:

a. Review your statements relating to the author contributions, and ensure you have specifically and accurately indicated the role(s) that these authors had in your study. These amendments should be made in the online form.

b. Confirm in your cover letter that you agree with the following statement, and we will change the online submission form on your behalf: 

“The funder provided support in the form of salaries for authors [insert relevant initials], but did not have any additional role in the study design, data collection and analysis, decision to publish, or preparation of the manuscript. The specific roles of these authors are articulated in the ‘author contributions’ section.

3. We noted in your submission details that a portion of your manuscript may have been presented or published elsewhere. "data were presented in the form of a virtual poster at the 2021 Society of General Internal Medicine Annual Meeting." Please clarify whether this conference proceeding was peer-reviewed and formally published. If this work was previously peer-reviewed and published, in the cover letter please provide the reason that this work does not constitute dual publication and should be included in the current manuscript.

5. Please ensure that you include a title page within your main document. We do appreciate that you have a title page document uploaded as a separate file, however, as per our author guidelines (http://journals.plos.org/plosone/s/submission-guidelines#loc-title-page) we do require this to be part of the manuscript file itself and not uploaded separately.

6. We note you have included a table to which you do not refer in the text of your manuscript. Please ensure that you refer to Table 4 in your text; if accepted, production will need this reference to link the reader to the Table.

Reviewers' comments:

Reviewer's Responses to Questions

**Comments to the Author**

1. Is the manuscript technically sound, and do the data support the conclusions?

Reviewer #1: Yes

Reviewer #2: Yes

Reviewer #3: Yes

2. Has the statistical analysis been performed appropriately and rigorously? 

Reviewer #1: Yes

Reviewer #2: I Don't Know

Reviewer #3: Yes

3. Have the authors made all data underlying the findings in their manuscript fully available?

Reviewer #1: Yes

Reviewer #2: No

Reviewer #3: No

4. Is the manuscript presented in an intelligible fashion and written in standard English?

Reviewer #1: Yes

Reviewer #2: Yes

Reviewer #3: Yes

5. Review Comments to the Author

Reviewer #1: I thought this study was well designed and well done. Here are some thoughts that I have for your study:

1. The sample size was ultimately quite small. Out of around 1300 trainees, only 33 participants took your survey. This makes it hard to make a substantial inference about the whole trainee experience during COVID-19 at your institution.

2. Negative change was the majority. This was an interesting finding that worthwhile sharing since it shows the stress that this population was under while also learning to be professionals.

3. You reference the "hidden curriculum" in your discussion without a citation. Can you either explain this or provide a reference that would explain it? I'm a medical trainee and understand this clearly, but it may not be clear to all readers.

4. I liked the discussion about why your learners didn't have a TL experience because of the pandemic. I think that in hindsight many years from now they may see this as a valuable learning experience.

5. Supplemental Table 1 - Do you have a reference for this information you can include?

Good luck with getting this published!

Reviewer #2: The manuscript titled “Transformative Learning of Medical Trainees during the COVID-19 Pandemic” utilized the Transformative Learning Survey approximately 6 months after the start of the pandemic in September and October 2020, to evaluate three process domains and four outcome subdomains, as well as coding two open-ended questions. Only 22 students (3.2%) and 11 residents/fellows (1.7%) responded, but the authors note they reached qualitative saturation. Of the four outcomes, Self-Awareness was the most prominent, and for the processes, Cognitive Rational and Social Critique were more prominent than Extrarational. Transformative learning through Social Critique was significantly more likely to be used by students than residents/fellows. For the qualitative analysis of the open-ended questions, negative changes were the most common and found significantly more in students than residents/fellows. Negative changes included education stressors, social isolation, and changes in daily life, while positive changes included more time with family and improved work-life balance.

A major concern is the very small sample size and how it can be representative of the pool. However, this is a very well-written and interesting manuscript, and can contribute to the field, particularly if a follow-up study is planned as only 8.4% of reported changes were defined as transformative and the authors believed that not enough time had elapsed since the start of the pandemic for adequate critical reflection, and the location of the study had a relatively low volume of COVID-19 positive patients. Listed below are suggestions and comments to help clarify some points and to potentially improve the utility of this manuscript.

Abstract:

• Background: The phrase “coronavirus disease 2019 pandemic” could be taken as 2019 pandemic, so it may be more clear to state “coronavirus disease 2019 (COVID-19) pandemic”

• Methods: Would be helpful to add the months of survey administration for context

• Results: perhaps listing some of the qualitative coding instead of just “negative changes were most common”

Methods:

• Study design, participants and procedure: It states that medical and PA students were grouped together for analysis given similarities in training structure – do both students have similar pre-requisites? What is the training structure? Perhaps a comparison of some demographics to show they are similar enough to combine?

• Data analysis, quantitative data: It states that if a significant difference was found in a process or outcome of TL between groups, then the same statistical tests were used to examine differences at the subdomain level – would it be possible that there were significant differences in the subdomain level that “canceled out” and wouldn’t be detectable at the higher levels?

Results:

• Demographic characteristics: PA students are divided into clinical and preclinical, but the medical students are not – please clarify. It would also be helpful to list the composition of the 22 students – PA vs medical, clinical vs preclinical.

• Demographic characteristics: The very poor response rate is unfortunate. Is there a way to compare the demographics of the participants vs the pool to be able to comment about it being representative?

• Quantitative results for transformative learning, Overall results: Supplementary Table 1 is stated, but it should refer to Supplementary Table 2.

• Quantitative results for transformative learning, Subgroup comparisons: The first set of numbers regarding Social Critique are not necessary because they are already in the table. I could not find the second set of numbers regarding sub-processes in any of the tables. Please list the results of data analysis in a table.

• Qualitative results for transformative learning, overall results: The inductive coding scheme does not seem to be Supplemental Table 1 – is it the first column of Table 4? And could you add p values to Table 4?

• Qualitative results for transformative learning, Positive change, shift or perception: The quotes at the top and middle of page 14 are not related to the sentences just prior to them that mention local politics and outdoor activities, respectively, so it doesn’t read as smooth.

Discussion:

• Implications for practice: A team-based fitness tracking program by the Department of Medicine was implemented after recruitment for the study was completed. Is there anything that was implemented because of the results of this study?

• Implications for practice: It states that not enough time had elapsed since the disorienting dilemma to truly allow for adequate critical reflection as the survey was completed relatively early in the pandemic when the large academic health center had a relatively low volume of COVID-19 positive patients – it would be interesting to repeat this study and compare – there may be an increase in transformative changes and a difference in Extrarational process

References:

• Unable to access reference #10 – the link did not work, and the name of the journal was not listed (Journal of Transformative Education).

Tables and Figures:

• Table 1: Please define graduate degrees – Masters, PhD and MDs? 5 of the students have graduate degrees.

• Table 2: The note implies that one can compare outcomes to processes, but I don’t think that is the intent. If so, perhaps the note should state, “All measures are grouped so that line items can be compared with other line items within the same heading”

• Table 3: Why are the housestaff boxes so much wider? Would be helpful to have asterisks on those that are significant.

• Table 4: Could you add p values?

• Supplemental Table 1: I am not sure this was referenced in the article. There were references to Supplemental Table 1 that did not seem to be correct – either Supplemental Table 2 or Table 4.

• Missing table? Under Quantitative results for transformative learning, Subgroup comparisons, there were a set of data regarding sub-processes that I couldn’t find.

Reviewer #3: Dear Authors,

It was a pleasure to review this manuscript. The authors have presented a timely and important topic. The following comments are attached to improve this study.

Authors presented ‘Each of these three processes of TL can be further differentiated into more specific subdomains’ suggest authors present a more clear description of subdomains.

Same for ‘Per institutional policy for education research implemented after COVID-19’ please elaborate more. Also, a brief description about how the learning and teaching opportunities were provided during the pandemic? What were the responsibilities of PA, staff, and others?

In the text, I could not see the transformative activities in the medicine program. How participants were informed about the different domains of TL? Please address more.

Sample size and sampling technique are missing.

The authors used a tool consisting of 90 items. It might be helpful for the authors to provide a brief overview of how they ensure the validation of responses? This might be one of the limitations of the low response rate. Please address more.

Few references are incorrect, please follow journal style.

Ethical Dimension - It clearly stated that the authors sought IRB approval, which is a good practice.

Once again, a valuable investigation but it may require some minor changes as discussed.

6. PLOS authors have the option to publish the peer review history of their article (what does this mean?). If published, this will include your full peer review and any attached files.

Reviewer #1: No

Reviewer #2: No

Reviewer #3: No

---

## [Author Response · Author response to Decision Letter 0]

19 Jul 2022

See attached word file with responses

---

## [Decision Letter · Decision Letter 1]

22 Aug 2022

PONE-D-21-19528R1Transformative learning of medical trainees during the COVID-19 pandemic: a cross sectional surveyPLOS ONE

Dear Dr. Vipler,

Thank you for submitting your manuscript to PLOS ONE. After careful consideration, we feel that it has merit but does not fully meet PLOS ONE’s publication criteria as it currently stands. Therefore, we invite you to submit a revised version of the manuscript that addresses the points raised during the review process. 1- Please clarify the study design inside the title. I believe its a mix method cross-sectional study, So, please change the title of manuscript.

Transformative learning of medical trainees during the COVID-19 pandemic: A mix method study.

Although you have explain the limitation of study, yet i request to justify your sample size inside the methodology.

2-Please prepare the checklist and attached it as supplementary file

https://journals.sagepub.com/doi/10.1177/1558689819875832

We look forward to receiving your revised manuscript.

Kind regards,

Muhammad Shahzad Aslam, Ph.D.,M.Phil., Pharm-D

Academic Editor

PLOS ONE

Journal Requirements:

Reviewers' comments:

Reviewer's Responses to Questions

**Comments to the Author**

1. If the authors have adequately addressed your comments raised in a previous round of review and you feel that this manuscript is now acceptable for publication, you may indicate that here to bypass the “Comments to the Author” section, enter your conflict of interest statement in the “Confidential to Editor” section, and submit your "Accept" recommendation.

Reviewer #2: (No Response)

2. Is the manuscript technically sound, and do the data support the conclusions?

Reviewer #2: Yes

3. Has the statistical analysis been performed appropriately and rigorously? 

Reviewer #2: Yes

4. Have the authors made all data underlying the findings in their manuscript fully available?

Reviewer #2: Yes

5. Is the manuscript presented in an intelligible fashion and written in standard English?

Reviewer #2: Yes

6. Review Comments to the Author

Reviewer #2: Thank you for the revisions of this manuscript. This reviewer (#2) also greatly appreciates the analysis done (to satisfy the reviewer’s curiosity) on the group differences for each measure in the other domains to find there was no statistically significant differences between the groups for the scales from the other process domains. The majority of comments were addressed except for the following:

It was suggested to add the months of survey administration and listing some of the qualitative coding within the abstract. The authors may have misunderstood this as their response was pointing out where they are located in the manuscript body. Perhaps there is a word limit to the abstract, but if possible, adding “We used the Transformative Learning Survey in September/October 2020…” would provide context for readers who are skimming the abstract to know that this occurred early in the pandemic.

7. PLOS authors have the option to publish the peer review history of their article (what does this mean?). If published, this will include your full peer review and any attached files.

Reviewer #2: No

---

## [Author Response · Author response to Decision Letter 1]

23 Aug 2022

PONE-D-21-19528R1

Transformative learning of medical trainees during the COVID-19 pandemic: a cross sectional survey

PLOS ONE

Dear Dr. Vipler,

Thank you for submitting your manuscript to PLOS ONE. After careful consideration, we feel that it has merit but does not fully meet PLOS ONE’s publication criteria as it currently stands. Therefore, we invite you to submit a revised version of the manuscript that addresses the points raised during the review process.

1- Please clarify the study design inside the title. I believe its a mix method cross-sectional study, So, please change the title of manuscript.

Transformative learning of medical trainees during the COVID-19 pandemic: A mix method study.

• Authors’ response: We have edited the title as such. 

Although you have explain the limitation of study, yet i request to justify your sample size inside the methodology.

• Authors’ response: We have included text in the methods section to address the editor’s concerns. This study was a pilot study (and funded as such) to examine transformative learning during COVID-19 in hopes of informing future work. As such, formal sample size calculations were not completed. Moreover, given recruiting restrictions placed on survey-based COVID-19 research at our institution, we were limited in further recruitment after the present sample size was achieved. Nevertheless, we believe this study holds valuable information that should be disseminated, and as stated, has already informed subsequent work, some of which we have already published. 

2-Please prepare the checklist and attached it as supplementary file

https://journals.sagepub.com/doi/10.1177/1558689819875832

• Authors’ response: We have included the completed checklist at the request of the editor. However, it appears this checklist was intended for methods papers within the domain of mixed methods (i.e., papers specifically designed with new methodologies). Thus, we do not believe all sections to be relevant for our article (mainly questions 16-20). Nevertheless, we have completed it to the best of our ability. 

We look forward to receiving your revised manuscript. 

Kind regards,

Muhammad Shahzad Aslam, Ph.D.,M.Phil., Pharm-D

Academic Editor

PLOS ONE

Journal Requirements:

Reviewers' comments:

Reviewer's Responses to Questions

Comments to the Author

1. If the authors have adequately addressed your comments raised in a previous round of review and you feel that this manuscript is now acceptable for publication, you may indicate that here to bypass the “Comments to the Author” section, enter your conflict of interest statement in the “Confidential to Editor” section, and submit your "Accept" recommendation.

Reviewer #2: (No Response)

2. Is the manuscript technically sound, and do the data support the conclusions?

Reviewer #2: Yes

3. Has the statistical analysis been performed appropriately and rigorously? 

Reviewer #2: Yes

4. Have the authors made all data underlying the findings in their manuscript fully available?

Reviewer #2: Yes

5. Is the manuscript presented in an intelligible fashion and written in standard English?

Reviewer #2: Yes

6. Review Comments to the Author

Reviewer #2: Thank you for the revisions of this manuscript. This reviewer (#2) also greatly appreciates the analysis done (to satisfy the reviewer’s curiosity) on the group differences for each measure in the other domains to find there was no statistically significant differences between the groups for the scales from the other process domains. The majority of comments were addressed except for the following:

It was suggested to add the months of survey administration and listing some of the qualitative coding within the abstract. The authors may have misunderstood this as their response was pointing out where they are located in the manuscript body. Perhaps there is a word limit to the abstract, but if possible, adding “We used the Transformative Learning Survey in September/October 2020…” would provide context for readers who are skimming the abstract to know that this occurred early in the pandemic.

• Authors’ response: We have added dates in the abstract as requested. We apologize for this oversight. With regard to including codes within the abstract, you are correct that we do not have codes per se. However, we do note our four aggregate codes (responses to the pandemic, dark grey in Table 4). Given that there are upwards of 14 individual codes detailed in table 4, we do not feel that we would be able to accomplish including these in a pithy manner in the abstract while keeping to the 300 word limit, which we have edited in this revision to meet exactly (excluding section headers) after the requested additions by reviewer #2. 

7. PLOS authors have the option to publish the peer review history of their article (what does this mean?). If published, this will include your full peer review and any attached files.

Do you want your identity to be public for this peer review? For information about this choice, including consent withdrawal, please see our Privacy Policy.

Reviewer #2: No

---

## [Editor Report · Decision Letter 2]

2 Sep 2022

Transformative learning of medical trainees during the COVID-19 pandemic: a mixed methods study

PONE-D-21-19528R2

Dear,

We’re pleased to inform you that your manuscript has been judged scientifically suitable for publication and will be formally accepted for publication once it meets all outstanding technical requirements.

Kind regards,

Muhammad Shahzad Aslam, Ph.D.,M.Phil., Pharm-D

Academic Editor

PLOS ONE
---

## [Editor Report · Acceptance letter]

8 Sep 2022

PONE-D-21-19528R2 

Transformative learning of medical trainees during the COVID-19 pandemic: a mixed methods study 

Dear Dr. Vipler:

I'm pleased to inform you that your manuscript has been deemed suitable for publication in PLOS ONE. Congratulations! Your manuscript is now with our production department. 

Kind regards, 

on behalf of

Dr. Muhammad Shahzad Aslam 

Academic Editor

PLOS ONE